# CodeMamba: Shifting from Target Semantics to Self-Supervised Background Manifold Learning for Singularity Detection in Infrared Sequences

**Jingwen Ma** [1]   **Xinpeng Zhang** [1]   **Fan Shi** [1]   **Xu Cheng** [1]   **Shengyong Chen** [1]

## Abstract

Multi-frame infrared small target detection suffers from extreme semantic paucity of targets and representation collapse due to overwhelming class imbalance, resulting in the persistent inability to accurately distinguish point-like targets from dynamic background clutter. To address these issues, we propose CodeMamba, a collaborative dual-stream framework that reframes this task as the complementary mechanisms of background manifold modeling and motion singularity capturing. The implicit stream emphasizes background regularity and anomaly localization, while the explicit stream focuses on motion consistency and spatiotemporal singularity. Finally, we design a Bayesian uncertainty-weighted fusion module that estimates the reliability of each stream by quantifying its observation noise. Extensive experiments on the IRDST and DAUB benchmarks demonstrate that CodeMamba not only outperforms existing methods but also achieves enhanced sensitivity to point-like targets.

## 1. Introduction

The task of time series anomaly detection aims to identify rare and irregular patterns that deviate from dominant normal behaviors, presenting a long-recognized challenge (Kim et al., 2025; Zhang et al., 2022). The difficulty of this task can be traced to two dilemmas: (i) the extreme imbalance between normal and anomalous observations, and (ii) the strong dependence of anomaly definitions on temporal context. Translating this research paradigm to the visual data introduces the additional complexity of jointly modeling spatial textures, temporal evolution, and dynamic background clutter. Moving Infrared Small Target Detection (MISTD) represents one of the most challenging instances of anomaly detection (Duan et al., 2024; Yang et al., 2025). In MISTD, targets occupy only sub-pixel regions with low contrast, which are frequently obscured by dynamic, cluttered backgrounds (Yuan et al., 2024; Zhu et al., 2025). These adverse factors render conventional detection paradigms fragile, leading to false alarms or missed detections (Zhang et al., 2023; Wu et al., 2025; Zhuang et al., 2025).

Recently, deep learning-based methods have focused on learning discriminative spatiotemporal feature representations from video sequences (Chen et al., 2024b; Duan et al., 2025). The mainstream strategies for motion modeling can be summarized as: (i) joint spatiotemporal filtering via convolutions (Liu et al., 2021); (ii) sequential state propagation via RNNs (Liu et al., 2025a; Chen et al., 2024a); and (iii) explicit motion estimation using optical flow (Wang et al., 2023) or feature alignment (Luo et al., 2026). More recent studies further enrich motion representations by incorporating auxiliary semantic supervision (Chen et al., 2025).

However, current modeling paradigms are often limited by a fundamental mismatch between their inductive biases and the inherent characteristics of targets.

- **Semantic Paucity within Standard Architectures.** Standard networks favor rich geometric structures like edges and textures, whereas infrared targets are inherently textureless and point-like. This misalignment hinders discriminative feature extraction, causing the network to frequently confuse stochastic noise with true targets.

- **Representation Collapse due to Extreme Class Imbalance.** Spatio-temporal backgrounds dominate the data volume, while targets are exceedingly sparse. The feature space overwhelmed by dominant background statistics collapses into a degenerate representation that fails to capture minority class distinctions.

To alleviate the aforementioned limitations, we propose CodeMamba, a theoretical framework that reformulates MISTD as a predictive learning task on a discrete latent

---

[1]Engineering Research Center of Learning-Based Intelligent System, Ministry of Education and the Key Laboratory of Computer Vision and System of Ministry of Education, Tianjin University of Technology, Tianjin, China. Correspondence to: Xinpeng Zhang <xpzhang@email.tjut.edu.cn>.

*Proceedings of the $43^{rd}$ International Conference on Machine Learning*, Seoul, South Korea. PMLR 306, 2026. Copyright 2026 by the author(s).

manifold. We break free from the conventional view that treats the background merely as a static low-rank component. This prompts us to establish a new perspective: the background as a complex neural dynamical system. Furthermore, the core challenge shifts from simplistic background suppression to learning the intrinsic topological manifold of the background from limited observations. This allows targets to be identified as outliers deviating from this manifold. Consequently, we propose two orthogonal inductive biases:

- **Manifold Regularity Bias:** Background exhibits high spatio-temporal redundancy, allowing compact representation via a learned discrete prototype space. Small targets, acting as irregular deviations, fail to fit these prototypes and naturally emerge as salient reconstruction residuals.

- **Motion Continuity Bias:** In contrast to the global and slowly varying dynamics of backgrounds, small targets are characterized by localized yet temporally consistent motion. Despite their spatial sparsity, their trajectories induce stable temporal gradients that persist across frames. Furthermore, they manifest as distinctive responses in both spatial and frequency domains. This observation motivates an explicit modeling of temporal gradient evolution in a joint spatial-frequency framework, which is critical for distinguishing them from non-stationary clutter.

Informed by these biases, we formulate CodeMamba as a collaborative dual-stream architecture. The implicit stream learns background regularity and anomaly localization via a Granularity-Aware Background Anomaly Perception Module (GBAPM), which maps background patterns onto the hierarchical coarse-to-fine codebooks in a self-supervised manner. Benefiting from the strong spatio-temporal redundancy of infrared backgrounds, the probabilistic vector quantization mechanism promotes compact background representation while preventing weak targets from being absorbed into continuous latent representations. Reconstruction residuals naturally form a prediction error map that serves as attention priors. The explicit stream introduces a Motion-Aware Hybrid Spatial–Frequency Mamba Module (MHSFM), which combines spatially selective scanning with frequency-domain saliency enhancement to extract discriminative motion singularity. Finally, a Bayesian Uncertainty-Weighted Fusion Module (BUWFM) integrates these complementary cues by adaptively reweighting each stream based on a reliability estimation, which is derived from quantifying its observation noise.

The main contributions of this work are summarized as follows:

- We propose a complementary dual-stream architecture

for MISTD by formulating the task as simultaneous background manifold modeling and motion singularity capturing.

- We propose a Granularity-Aware Background Anomaly Perception Module (GBAPM) that leverages hierarchical probabilistic vector quantization (PVQ) to distill stable and accurate background prototypes, thereby amplifying the saliency of small target regions.

- We design a Motion-Aware Hybrid Spatial-Frequency Mamba Module (MHSFM) that integrates spatial selective scanning with global frequency-domain modeling to capture subtle motion trajectories effectively, thereby ensuring the model remains computationally efficient.

- We propose a Bayesian Uncertainty-Weighted Fusion Module (BUWFM) that adaptively weights each stream by quantifying its observation noise.

## 2. Related work

### 2.1. Multi-Frame Small Target Detection

Early studies predominantly treated the background as a low-rank component, with targets modeled as sparse perturbations in image patch sequences (Gao et al., 2013; Lin et al., 2010; Zhang & Peng, 2019). However, morphological background clutter such as cloud edges and sea waves often violate the low-rank assumption, resulting in frequent false alarms (Zhu et al., 2024b). Subsequently, deep learning-based methods shifted the paradigm of MISTD toward end-to-end learning of spatiotemporal features.

One prevalent approach employs 3D or temporal convolutions to jointly model consecutive frames and capture local spatiotemporal patterns. For example, Ma et al. (Ma et al., 2025) employed 3D depth-wise separable convolutions to construct a weight-shared temporal feature network. Furthermore, several methods incorporate optical flow, temporal attention, or memory mechanisms to explicitly model target motion or long-term temporal dependencies. For instance, MSTCNet (Cui et al., 2025) combined CNNs and ConvLSTMs, leveraging ConvLSTM to extract motion information from deep features and thereby enhance generalization. However, these methods directly treated weak targets as a conventional semantic category, leading to overfitting.

### 2.2. Visual State Space Model

Early works such as ViS4mer (Islam & Bertasius, 2022) embedded State Space Models (SSMs) into the feed-forward blocks of Vision Transformers to strengthen long-range dependency modeling. Recently, Vision Mamba (Vim) (Zhu et al., 2024a) adopted bidirectional SSMs to establish a

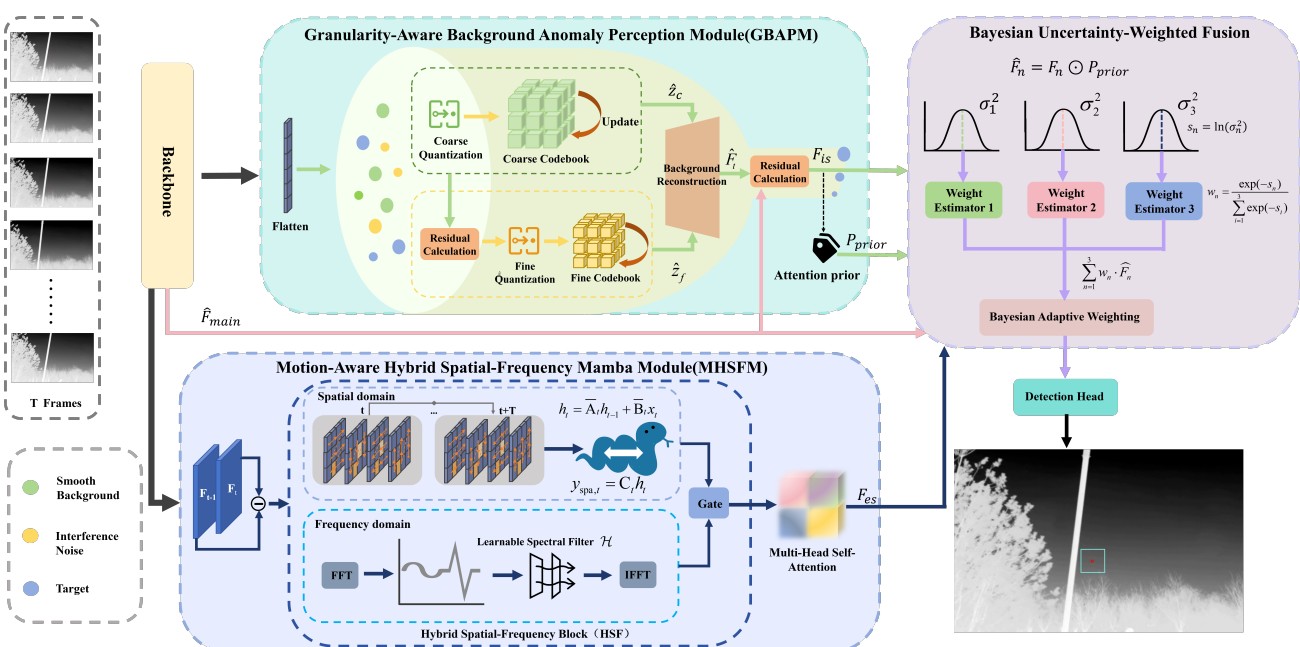

*Figure 1.* The CodeMamba framework consists of three core components: (1) a GBAPM module for modeling stable background manifolds; (2) a MHSFM module for capturing transient dynamics; and (3) a BUWFM module that adaptively weights features based on prediction variance.

purely SSM-based visual backbone. Vim demonstrated highly competitive performance on large-scale image recognition benchmarks, while significantly reducing memory consumption. This linear-time efficiency advantage was subsequently extended to the video domain. Furthermore, Video Mamba (Li et al., 2024) introduced selective state space scanning for spatiotemporal modeling to efficiently aggregate long-range temporal dependencies.

## 3. Method

We propose CodeMamba, a collaborative dual-stream architecture designed to disentangle the spatiotemporal modeling of infrared small targets into two orthogonal tasks: *Implicit Manifold Screening* and *Explicit Motion Extraction*. As illustrated in Figure 1, we select CSPDarknet (Bochkovskiy et al., 2020) as the backbone to extract a sequence of multi-scale feature maps $F_{main}$ at time $t$, which are then fed into two parallel pathways. The implicit stream projects $F_{main}$ onto two hierarchical learnable codebooks through a Probabilistic Vector Quantization (PVQ) algorithm, which synthesizes a background prior $\hat{F}_t$ together with an attention prior $P_{prior}$. Simultaneously, the explicit stream utilizes a hybrid spatial-frequency mechanism to adaptively model temporal residuals and effectively characterize motion differences in both the spatial and frequency domains, thereby accurately extracting motion singularities. The Mamba-based design is

introduced in the explicit stream to efficiently capture long-range temporal dependencies and dynamic state transitions across frames with linear computational complexity. Finally, a Bayesian uncertainty-weighted fusion module integrates these complementary cues for final detection.

### 3.1. Implicit stream: Granularity-Aware Background Anomaly Perception Module

Given the extremely limited semantic cues of point-like targets, we advocate for a new perspective based on self-supervised background manifold identification. At its core is the Granularity-Aware Background Anomaly Perception Module (GBAPM), which executes a manifold-sieving strategy to identify target regions that deviate from the background. Specifically, this module employs a hierarchical PVQ mechanism that maps complex background textures onto coarse-to-fine codebooks with a fixed size to distill a compact representation by capturing their latent statistical regularities. Small targets manifest as deviations from the prototype distribution of background. Consequently, infrared small targets are identified as abnormal signals to yield a self-supervised attention prior, which can be discriminated from background in the absence of rich semantic cues.

Firstly, we design a coarse-grained codebook $\mathcal{C}_c$ that can effectively capture low-frequency structural priors, such as

large cloud masses. However, using hard vector quantization may introduce blocky artifacts at transitional boundaries when modeling continuous thermal fields. To address this issue, we introduce a weighted mixture via the Gumbel-Softmax mechanism (Jang et al., 2017). Let $\pi_{c,k}$ denote the probability of a latent feature $z$ that is assigned to the coarse codeword $e_{c,k}$, which is computed as follows:

$$\pi_{c,k} = \frac{\exp(-(\|z - e_{c,k}\|_2^2 + g_k)/\tau)}{\sum_k \exp(-(\|z - e_{c,k}\|_2^2 + g_k)/\tau)} \qquad (1)$$

where $\tau$ is the temperature and $k \in 1, \ldots, K$ denotes the codeword index. The term $\|z - e_{c,k}\|_2^2$ measures the similarity between feature $z$ and background prototype $e_{c,k}$ in squared $L_2$ distance. $g_1, \ldots, g_K$ are i.i.d. samples drawn from Gumbel$(0, 1)$. The relaxation degree is controlled by a temperature parameter $\tau > 0$, which governs the transition to a one-hot categorical encoding as $\tau \to 0$. The denominator sums over all codewords $j \in \{1, \ldots, K\}$ to construct a normalized probability manifold of background. Subsequently, the coarse component $\hat{z}_c$ is synthesized as:

$$\hat{z}_c = \sum_k \pi_{c,k} e_{c,k} \qquad (2)$$

Nevertheless, a singular coarse codebook is limited in its capacity to capture the nuances within complex backgrounds. The remaining residual information $R$ preserves crucial finer details. Therefore, we construct another fine-grained codebook $\mathcal{C}_f$ to distill the residual information $R$ through the same Gumbel-Softmax relaxation, yielding a complementary component $\hat{z}_f$.

$$R = z - \hat{z}_c \qquad (3)$$

$$\pi_{f,k} = \frac{\exp(-(\|R - e_{f,k}\|2^2 + g_k)/\tau)}{\sum_j \exp(-(\|R - e_{f,j}\|_2^2 + g_j)/\tau)} \qquad (4)$$

$$\hat{z}_f = \sum_k \pi_{f,k} e_{f,k} \qquad (5)$$

Finally, the complete background representation is obtained by summing the coarse and fine components:

$$\hat{z}_{bg} = \hat{z}_c + \hat{z}_f \qquad (6)$$

As a result, the complete background reconstruction feature $\hat{F}_t$ is generated via a decoder. Furthermore, we calculate the reconstruction residual between $F_{main}$ and $\hat{F}_t$ to yield a kind of attention priors $P_{prior}$, which guide the network to focus on target regions, as shown in Figure 2. Benefiting from iterative background manifold optimization, the generated attention priors provide progressively refined spatial supervision,

$$F_{is} = |F_{main} - \hat{F}_t| \qquad (7)$$

$$P_{prior} = \phi(F_{is}) \qquad (8)$$

where $\phi$ represents the Sigmoid function. To adaptively capture the abnormal distribution, we employ an exponential moving average (EMA) updating strategy (Tarvainen & Valpola, 2017). At each training iteration $t$, we update the codewords $e_{l,k}, l \in \{c, f\}$, based on the spatial statistics aggregated over the entire feature map. Firstly, for $k$-th codeword $e_{l,k}$ in codebook $l$, we compute its effective assignment mass $N_{l,k}^{(t)}$ by accumulating the Gumbel-Softmax probabilities $\pi_{l,k,i}$ across all spatial locations $i \in \{1, \ldots, H \times W\}$:

$$N_{l,k}^{(t)} = \gamma N_{l,k}^{(t-1)} + (1 - \gamma) \sum_{i=1}^{H \times W} \pi_{l,k,i} \qquad (9)$$

where $\gamma \in [0, 1)$ is the decay factor controlling the update speed, $H$ and $W$ denote the height and width of feature maps. Simultaneously, $M_{l,k}^{(t)}$ is also updated as follows:

$$M_{l,k}^{(t)} = \gamma M_{l,k}^{(t-1)} + (1 - \gamma) \sum_{i=1}^{H \times W} \pi_{l,k,i} z_{l,i} \qquad (10)$$

where the input $z_{l,i}$ of each codebook $l \in \{c, f\}$ is defined as follows:

$$z_{l,i} = \begin{cases} z_i, & \text{if } l = c \\ r_i, & \text{if } l = f \end{cases} \qquad (11)$$

Secondly, the codeword $e_{l,k}^{(t)}$ is updated by normalizing the accumulated features:

$$e_{l,k}^{(t)} = \frac{M_{l,k}^{(t)}}{N_{l,k}^{(t)} + \epsilon} \qquad (12)$$

where $\epsilon$ is a small constant for numerical stability.

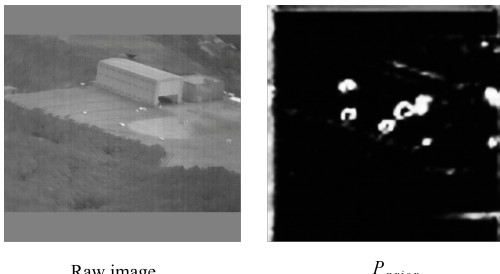

Raw image           $P_{prior}$

*Figure 2.* Visualizations of attention priors.

Owing to the extreme sparsity of small targets, their marginal contributions to the spatial sums $\sum_i \pi_{l,k,i} z_{l,i}$ are almost overwhelmed by the abundant background regions. Consequently, the codewords are statistically driven to quantize only the dominant background manifold in a discrete latent space. The reconstructed background $\hat{F}_t$ can exclude anomalous target signatures, allowing the attention prior $P_{prior}$ to accurately emphasize the target regions.

## 3.2. Explicit stream: Motion-Aware Hybrid Spatial-Frequency Mamba Module

Although GBAPM provides a robust manifold-sieving mechanism to isolate potential small target regions, it encounters significant challenges when targets exhibit extremely low intensity or thermal distributions that closely resemble the local background. In addition, the severe data imbalance leads to representation collapse and feature drift, particularly in dynamic scenes. To address these issues, we design a Motion-Aware Hybrid Spatial–Frequency Mamba module (MHSFM) to explicitly capture motion singularities of infrared small targets. This module incorporates a Hybrid Spatial-Frequency block (HSF) that integrates spatial selective scanning with global frequency-domain modeling to effectively disentangle target motion from background dynamics. Specifically, we first compute temporal difference $D_t$ between adjacent frames to eliminate redundant background information. Subsequently, we project $D_t$ into a latent sequence $x \in \mathbb{R}^{L \times D}$, which is processed in parallel along two separate branches: spatial selective scanning and global frequency-domain modeling.

In the spatial domain, temperature typically varies smoothly, such as drifting clouds or ocean waves without sharp intensity changes. Furthermore, the heat diffusion also follows a regular spatiotemporal pattern over short intervals. Motivated by these physical properties, the spatial branch focuses on modeling local thermal continuity and short-range temporal evolution through selective state space model. At each step $t \in \{1, \ldots, L\}$, the hidden state $h_t$ is updated and then used to compute the spatial output $y_{\text{spa},t}$:

$$h_t = \bar{\mathbf{A}}_t h_{t-1} + \bar{\mathbf{B}}_t x_t, \quad y_{\text{spa},t} = \mathbf{C}_t h_t, \qquad (13)$$

where the system transition matrices $(\bar{\mathbf{A}}_t, \bar{\mathbf{B}}_t, \mathbf{C}_t)$ are parameterized by the current input $x_t$. Specifically, $\bar{\mathbf{A}}_t = \exp(\Delta_t \mathbf{A})$, where $\Delta_t$ and $(\bar{\mathbf{B}}_t, \mathbf{C}_t)$ are obtained through linear projections of $x_t$. Regions with different motion conditions update their states differently over time. Therefore, the spatial branch can adapt to changes in context while maintaining fine spatial information.

In the frequency domain, background motion, such as the drifting of clouds or undulating of waves are typically characterized by low-frequency components. While targets act as broadband impulses. Therefore, we design a Spectral Saliency block to highlight targets that utilizes the 2D Fast Fourier Transform (FFT) to transform $x_t$ into frequency domain:

$$X_t(u, v) = \mathcal{F}(x_t) = \sum_{h=0}^{H-1} \sum_{w=0}^{W-1} x_t(h, w) e^{-j2\pi \left( \frac{uh}{H} + \frac{vw}{W} \right)} \tag{14}$$

where $(u, v)$ denote spatial coordinates in the frequency domain. Furthermore, a learnable spectral filter $\mathcal{H}$ (Rao et al.,

2021) is applied to suppress low-frequency components:

$$\tilde{X}_t = X_t \odot \mathcal{H} \tag{15}$$

The enhanced motion features $y_{\text{spec},t}$ are then reconstructed via the Inverse FFT:

$$y_{\text{spec},t} = \mathcal{F}^{-1}\left( \tilde{X}_t \right) \tag{16}$$

Finally, a unidirectional gating mechanism is employed to fuse $y_{\text{spa},t}$ and $y_{\text{spec},t}$:

$$y_{\text{out}} = \text{Gate}(y_{\text{spa},t},\, y_{\text{spec},t}) \tag{17}$$

To model the spatiotemporal consistency of targets, we develop a bidirectional HSF strategy. Two parallel HSFs are designed to scan the motion tensor sequence in opposite temporal directions to obtain full-range motion context:

$$M_t^{dyn} = \frac{1}{2} \left( \overrightarrow{\text{HSF}}(x_t) + \overleftarrow{\text{HSF}}(x_t) \right) \tag{18}$$

where $\overrightarrow{\text{HSF}}$ aggregates information from past frames to capture the motion changes over time, while $\overleftarrow{\text{HSF}}$ incorporates information from future frames to verify implicit temporal consistency. Finally, we utilize the multi-head self-attention mechanism to obtain the final result $F_{es}$. This module can specifically enhance target signals that exhibit stable and coherent motion across frame dimensions.

### 3.3. Bayesian Uncertainty-Weighted Fusion Module

A common limitation in conventional fusion strategies is the presumption of uniform reliability among all contributing branches. For MISTD, however, data streams are highly susceptible to interference from severe camera jitter or sensor noise. To address this issue, we design a Bayesian uncertainty-weighted fusion module that adaptively calibrates the contribution of each feature source. The fusion integrates four heterogeneous features derived from the dual processing streams: backbone features $F_{main}$; explicit stream features $F_{es}$; implicit stream features $F_{is}$; attention priors $P_{prior}$. Firstly, we perform element-wise multiplication with the multi-source features using an attention prior map $P_{prior}$ to highlight target-related regions:

$$\hat{F}_n = F_n \odot P_{\text{prior}}, \quad n \in \{1, 2, 3\}. \tag{19}$$

Following Bayesian deep learning principles (Kendall & Gal, 2017), $\hat{F}_n$ can be modeled as a Gaussian distribution:

$$y_n \sim \mathcal{N}(\hat{f}_n, \sigma_n^2) \tag{20}$$

where $\hat{f}_n$ denotes the mean value. The variance $\sigma_n^2$ represents the observation noise which is defined as a measure of stochastic uncertainty. A higher value of $\sigma_n^2$ corresponds

to lower reliability. Therefore, we design a weight estimator to deduce variance of each feature source using Global Average Pooling (GAP) followed by a $1 \times 1$ convolution. The result is considered as the log-variance of each feature, $s_n = \ln \sigma_n^2$.

According to the principle of inverse-variance weighting, the optimal fusion of independent noisy observations assigns lower weights to sources with higher uncertainty. Therefore, we apply a Softmax function to transform the predicted log-variances into normalized weights, which are then used for weighted fusion:

$$w_n = \frac{\exp(-s_n)}{\sum_{i=1}^{3} \exp(-s_i)} \quad (21)$$

The fused feature representation $F_{\text{fused}}$ is then obtained as a weighted summation:

$$F_{\text{fused}} = \sum_{n=1}^{3} w_n \cdot \hat{F}_n \quad (22)$$

This module perceives the uncertainty arising from environmental variations to guide an adaptive and robust feature fusion strategy.

### 3.4. Loss Function

The total loss $\mathcal{L}_{total}$ is a weighted sum of the explicit detection loss $\mathcal{L}_{det}$ and the implicit generative loss $\mathcal{L}_{imp}$, with a coefficient $\lambda$ controlling their trade-off:

$$\mathcal{L}_{total} = \mathcal{L}_{det} + \lambda \mathcal{L}_{imp} \quad (23)$$

For the detection head, we adopt the decoupled loss formulation from YOLOX (Ge et al., 2021) to supervise the extraction of motion singularities:

$$\mathcal{L}_{det} = \mathcal{L}_{cls} + \Theta_{reg} \mathcal{L}_{reg} + \Theta_{obj} \mathcal{L}_{obj} \quad (24)$$

where $\mathcal{L}_{cls}$ is the Binary Cross Entropy (BCE) loss for classification, $\mathcal{L}_{reg}$ is the IoU loss for bounding box regression, and $\mathcal{L}_{obj}$ denotes Binary Cross Entropy (BCE) loss to calculate object confidence.

The implicit stream is optimized via a self-supervised task. To encourage the hierarchical codebooks $(\mathcal{C}_c, \mathcal{C}_f)$ to capture the dominant background statistics within a discrete latent representation, we express the implicit flow loss as:

$$\mathcal{L}_{imp} = \mathcal{L}_{recon} + \alpha \mathcal{L}_{commit} + \beta \mathcal{L}_{entropy} \quad (25)$$

$\mathcal{L}_{recon}$ represents reconstruction loss:

$$\mathcal{L}_{recon} = \|\text{sg}(F_{main}) - \hat{F}_t\|_2^2 \quad (26)$$

where $\text{sg}(\cdot)$ denotes the stop-gradient operator. We minimize the $L_2$ distance between the input feature $F_{main}$ and the hierarchical synthesized background $\hat{F}_t$ to ensure that the manifold reliably represents generalized background. $\mathcal{L}_{commit}$ is the commitment loss that prevents the encoded features from drifting away from codebooks.

$$\mathcal{L}_{commit} = \|F_{main} - \text{sg}(\hat{F}_t)\|_2^2 \quad (27)$$

$\mathcal{L}_{entropy}$ denotes the sparsity entropy loss. To adequately model the background in dynamic scenes, we apply an entropy regularization to the Gumbel-Softmax assignment probabilities $\pi$.

$$\mathcal{L}_{entropy} = \sum_{l \in \{c,f\}} \left( -\frac{1}{H \times W} \sum_x \sum_k \pi_{l,k}^{(x)} \log(\pi_{l,k}^{(x)} + \epsilon) \right) \quad (28)$$

## 4. Experiments

### 4.1. Experimental Setup

**Datasets:** We conduct experiments on two benchmark datasets for MISTD, namely DAUB (Hui et al., 2019) and IRDST (Sun et al., 2023). The DAUB dataset contains ten video sequences for training, comprising 8,983 frames, while the test split includes seven sequences with 4,795 frames. The IRDST dataset is larger in scale, with 42 training videos totaling 20,398 frames and 43 test videos containing 20,258 frames.

**Implementation Details:** The proposed network is implemented using PyTorch and trained on an NVIDIA RTX 4090 GPU. We adopt the AdamW optimizer with an initial learning rate of $1 \times 10^{-3}$ and a cosine annealing schedule. The batch size is set to 4, and the input resolution is $512 \times 512 \times 5$ (frames). For hyperparameters, we set the balancing coefficient $\lambda = 0.1$, the commitment weight $\alpha = 1.0$, and the entropy weight $\beta = 0.1$.

**Evaluation Metrics:** Quantitative evaluation of our framework is conducted through a multi-dimensional metric framework. We utilize Precision (Pr) and Recall (Re) to characterize the trade-off between sensitivity and false-alarm suppression, supplemented by the F1-score to represent their harmonic mean. To further capture the overall robustness of the detection results, the mean Average Precision (mAP50) is adopted, measuring the precision-recall area under a standard IoU threshold of 0.5.

### 4.2. Quantitative Analysis

#### 4.2.1. COMPARISONS WITH STATE-OF-THE-ARTS

Our evaluation includes nine representative methods from two categories. Single-frame detectors include AGPCNet and SCTransNet. Sequence-based models comprise TSI-Net, MICPL, TMP, MoPKL, STME, Tridos, and SSTNet. For a fair comparison, all methods are retrained on the

*Table 1.* Quantitative comparison on IRDST and DAUB datasets. **Bold** indicates the best performance.

| Methods | IRDST | | | | DAUB | | | |
|---|---|---|---|---|---|---|---|---|
| | $mAP50\%$ | $Pr\%$ | $Re\%$ | $F1\%$ | $mAP50\%$ | $Pr\%$ | $Re\%$ | $F1\%$ |
| AGPCNet (Zhang et al., 2023) | 59.21 | 79.47 | 75.51 | 77.44 | 76.72 | 82.29 | 94.43 | 87.95 |
| SCTransNet (Yuan et al., 2024) | 64.35 | 82.10 | 78.45 | 80.23 | 85.60 | 88.50 | 95.12 | 91.69 |
| TSI-Net (Zhuang et al., 2025) | 68.42 | 83.15 | 80.30 | 82.18 | 93.46 | 90.21 | 95.56 | 93.37 |
| MICPL (Chen et al., 2024b) | 72.80 | 87.20 | 83.56 | 85.34 | 95.85 | 98.15 | 98.22 | 98.18 |
| TMP (Zhu et al., 2024b) | 70.01 | 86.55 | 81.23 | 83.89 | 93.50 | 96.80 | 97.10 | 96.95 |
| MoPKL (Chen et al., 2025) | 74.54 | 89.04 | 84.74 | 86.84 | 96.35 | 98.60 | 98.50 | 98.55 |
| STME (Peng et al., 2025) | 73.40 | 87.78 | 84.22 | 85.96 | 96.56 | 98.82 | 98.62 | 98.72 |
| MIST-Net (Zhang et al., 2026) | 73.49 | - | - | 87.00 | 95.62 | - | - | 97.30 |
| GST-Det (Liu et al., 2025b) | - | - | - | - | 97.31 | - | - | - |
| Tridos (Duan et al., 2024) | 73.72 | 84.49 | 89.35 | 86.85 | **97.80** | **99.20** | 99.67 | 99.43 |
| SSTNet (Chen et al., 2024a) | 71.55 | 88.56 | 81.92 | 85.11 | 95.59 | 98.08 | 98.10 | 98.09 |
| **Ours** | **78.24** | **90.85** | **90.40** | **90.62** | 97.42 | 99.10 | **99.75** | **99.42** |

IRDST and DAUB datasets using the default hyperparameters provided in their official implementations. Table 1 comprehensively evaluates the performance of our method on the IRDST and DAUB datasets compared to the nine state-of-the-art methods.

After examining the dataset characteristics, we observe that IRDST poses a more challenging detection scenario due to its highly dynamic background clutter and the frequent proximity of targets to high-intensity background structures with similar appearances.

On the IRDST dataset, our method achieves a Pr of 90.85%, with gains of 1.81% and 2.29% over the state-of-the-art MoPKL and SSTNet, respectively. This precision advantage is primarily attributed to the manifold filtering mechanism employed in the implicit flow. It purifies the feature space by excising dynamic background noise, resulting in a lower false alarm rate. Although SSTNet integrates temporal contexts via ConvLSTM, it often struggles to distinguish between background and target motion in high-dynamic scenes. While MoPKL leverages language-based motion priors which are directly extracted from images, it fails in scenes with complex background textures or persistent dynamic interference. To adapt the non-stationary scenes, TSI-Net designs a semantic interaction strategy, however, its temporal attention mechanism may struggle to accurately identify subtle targets. MICPL and TMP achieve competitive accuracy through motion-visual disentangling and spatially assisted temporal motion perception. However, their recall rates (83.56% and 81.23%, respectively) indicate that a certain proportion of real targets are still not detected in complex scenarios. In contrast, our method achieves recall improvements of 6.84% and 9.17%. This is attributed to the hybrid spatial-frequency domain design in explicit stream, which not only captures the motion singularity of weak targets using frequency domain transformations but also enhances the temporal consistency through bidirectional motion scanning. In addition, our method outperforms Tri-

dos's three-domain parallel fusion method by 3.77% in F1 score. This indicates that the three modules form an effective collaborative mechanism, significantly reducing missed detections while maintaining high detection accuracy. Furthermore, we utilize Precision-recall (PR) curves to further evaluate detection performance under different confidence thresholds, as shown in Figure 3. Our method sustains superior precision at equivalent recall rates, demonstrating a consistently favorable balance across the entire recall range.

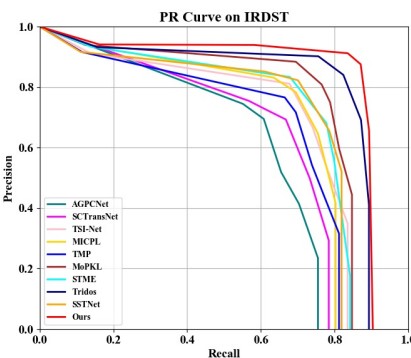

(a) IRDST dataset

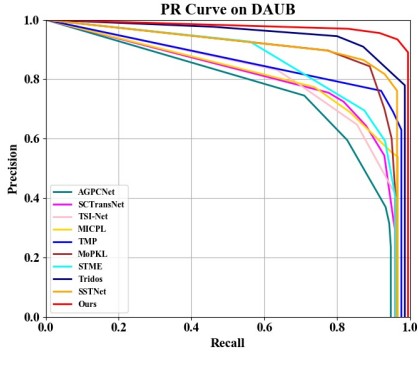

(b) DAUB dataset

*Figure 3.* PR curves on the DAUB and IRDST datasets.

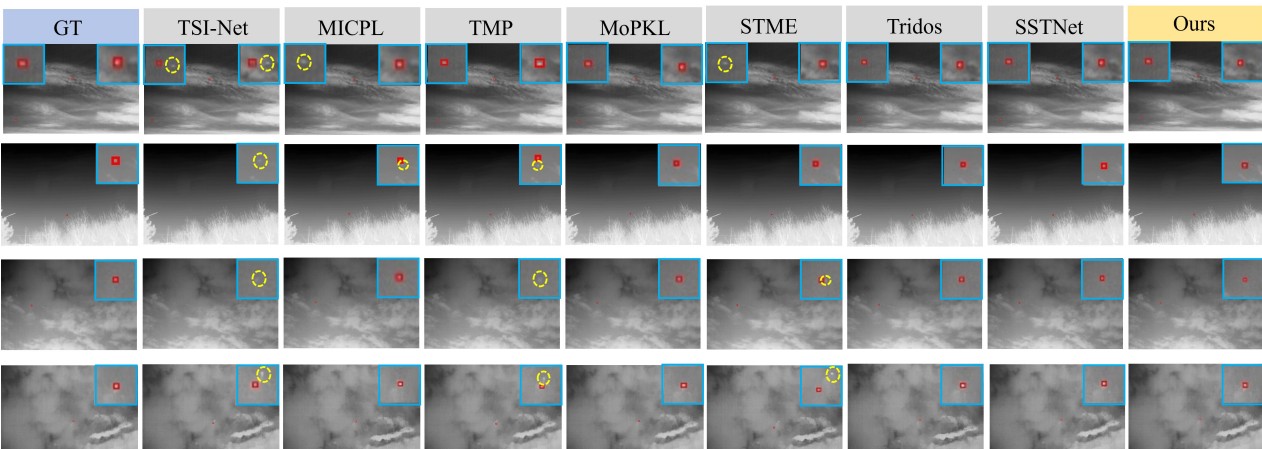

*Figure 4.* Qualitative detection results of our CodeMamba method compared to several representative SOTA methods.

*Table 2.* Ablation of dual-stream components and Bayesian fusion on IRDST.

| GBAPM | MHSFM | Bayesian | $mAP50\%$ | $Pr\%$ | $Re\%$ | $F1\%$ |
|---|---|---|---|---|---|---|
| ✓ | – | – | 76.31 | 88.84 | 84.59 | 86.66 |
| – | ✓ | – | 76.86 | 87.18 | 85.46 | 86.31 |
| ✓ | ✓ | – | 77.35 | 89.29 | 87.30 | 88.28 |
| ✓ | ✓ | ✓ | **78.24** | **90.85** | **90.40** | **90.62** |

*Table 3.* Ablation of dual-stream components and Bayesian fusion on DAUB.

| GBAPM | MHSFM | Bayesian | $mAP50\%$ | $Pr\%$ | $Re\%$ | $F1\%$ |
|---|---|---|---|---|---|---|
| ✓ | – | – | 96.32 | 98.41 | 98.31 | 98.36 |
| – | ✓ | – | 96.05 | 98.67 | 98.05 | 98.36 |
| ✓ | ✓ | – | 96.74 | 99.08 | 98.74 | 98.91 |
| ✓ | ✓ | ✓ | **97.42** | **99.10** | **99.75** | **99.42** |

### 4.2.2. ABLATION STUDY

Ablation experiments quantitatively reveal the unique contributions of each component in the two-stream architecture. As listed in Table 2 and Table 3, GBAPM acts as a manifold filter to accurately reconstruct dominant background patterns. Consequently, regions where small targets deviate from the learned background manifold are highlighted, effectively suppressing background-induced false alarms and enhancing detection precision. Furthermore, MHSFM captures the singularity and consistency of target motion to detect weak targets that are difficult to distinguish by pure spatial cues, resulting in substantially higher recall. In addition, the method adaptively adjusts the weighting of each feature source by quantifying its observation noise, resulting in robust localization and reliable detection for small targets.

*Table 4.* Effectiveness of spatial and frequency paths in HSF.

| Spatial Path | Frequency Path | $mAP50\%$ | $Pr\%$ | $Re\%$ | $F1\%$ |
|---|---|---|---|---|---|
| ✓ | – | 75.80 | 85.87 | 82.94 | 84.37 |
| – | ✓ | 73.33 | 86.29 | 81.85 | 84.03 |
| ✓ | ✓ | **78.24** | **90.85** | **90.40** | **90.62** |

We further investigate the respective contributions of the spatial domain and frequency domain within the HSF module through component-wise ablation. The ablation results are summarized in Table 4. The model employs two synergistic pathways: the spatial domain enhances recall by capturing local textures and thermal details, while the frequency domain improves precision via global spectral filtering that suppresses background clutter.

*Table 5.* Impact of scanning direction in bidirectional HSF.

| Direction | $mAP50\%$ | $Pr\%$ | $Re\%$ | $F1\%$ |
|---|---|---|---|---|
| Forward only | 72.15 | 83.32 | 81.61 | 82.46 |
| Backward only | 69.40 | 81.41 | 79.27 | 80.33 |
| **Bidirectional** | **78.24** | **90.85** | **90.40** | **90.62** |

As listed in Table 5, we analyze the impact of scanning direction in the bidirectional HSF strategy. The relatively low $Re$ under unidirectional scanning (forward or backward) is due to the vulnerability of a single temporal perspective to trajectory interruptions caused by occlusion or weak intensity, leading to missed detections. In contrast, the bidirectional mechanism achieves a recall of 90.4% by leveraging future information from the backward scan to recover targets in low-signal frames, effectively restoring broken trajectories. Meanwhile, random thermal noise typi-

cally lacks bidirectional temporal coherence. Through the bidirectional consistency verification, our model suppresses spurious disturbers that manifest as unidirectional artifacts in the temporal sequence.

### 4.3. Qualitative Analysis

Figure 4 illustrates the qualitative detection results of our CodeMamba compared to several representative SOTA methods in typical challenging scenarios. Due to the lack of effective background manifold modeling, bounding boxes generated by methods such as TSI-Net and TMP often drift into background clutter. In contrast, CodeMamba sieves out interference through an implicit flow manifold sieve mechanism. By integrating spatial and frequency domains within an explicit framework, our method achieves accurate localization of weak targets even amidst intense noise.

## 5. Conclusion

In this paper, we propose CodeMamba, a novel complementary dual-stream architecture for singularity detection in infrared sequences. Firstly, the GBAPM distills a consistent and stable background manifold via hierarchical quantization, effectively reframing targets as statistical deviations. Secondly, the MHSFM captures the motion singularity and trajectory consistency of targets by jointly modeling spatial textures and spectral components. Finally, the BUWFM adaptively weights multi-stream features for robust fusion through estimating their uncertainty. Future work will explore extending this self-supervised paradigm to cross-modal tasks, such as infrared-visible fusion and multispectral detection.

## Acknowledgments

This work was supported by the National Natural Science Foundation of China under Grant 62020106004, Grant T2422015 and Grant 62272342. The work of Xu Cheng was support by the Marie Skłodowska-Curie Actions (MSCA) under Project 101111188.

## Impact Statement

This paper presents work whose goal is to advance the field of computer vision. There are many potential societal consequences of our work, none of which we feel must be specifically highlighted here.

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

# A. Appendix

## A.1. Impact of Codebook Size $K$

Table 6 reports the impact of the codebook size $K$ in GBAPM on detection performance. When $K$ is too small, the codebook lacks sufficient capacity to represent diverse background patterns. Increasing $K$ from 128 to 512 leads to consistent gains across all evaluation metrics. This trend indicates that a larger codebook capacity promotes a more effective representation of the background manifold to discriminate small targets. However, further increasing the codebook size to $K = 1024$ leads to significant performance degradation. This is because an overly large codebook dilutes the statistical representation, as each codeword is assigned too few and unrepresentative features, leading to unreliable prototype estimation. Experimental results demonstrate that our network performs best at $K = 512$.

*Table 6.* Ablation study on Codebook Size $K$ in GBAPM.

| Codebook Size ($K$) | $mAP50\%$ | $Pr\%$ | $Re\%$ | $F1\%$ |
|---|---|---|---|---|
| $K = 128$ | 69.89 | 82.27 | 81.15 | 81.71 |
| $K = 256$ | 74.23 | 85.56 | 83.92 | 84.73 |
| $K = 512$ | **78.24** | **90.85** | **90.40** | **90.62** |
| $K = 1024$ | 74.38 | 86.31 | 84.91 | 85.60 |

## A.2. Sensitivity Analysis of Hyperparameter

Table 7 investigates the sensitivity of hyperparameter $\lambda$, which controls the contribution of the implicit self-supervised loss. From the results in Table 7, our method yields peak performance when $\lambda = 0.1$. This indicates that moderate self-supervised regularization can effectively enhance background modeling while preserving weak targets. Conversely, a larger $\lambda$ over-regularizes the model toward the background manifold at the expense of suppressing or assimilating the subtle signatures of anomalous targets. This leads to increased missed detections, as evidenced by the noticeable decline in recall.

*Table 7.* Sensitivity analysis of hyperparameter $\lambda$.

| Values of $\lambda$ | $mAP50\%$ | $Pr\%$ | $Re\%$ | $F1\%$ |
|---|---|---|---|---|
| $\lambda = 0.1$ | **78.24** | **90.85** | **90.40** | **90.62** |
| $\lambda = 0.5$ | 77.48 | 89.37 | 89.12 | 89.24 |
| $\lambda = 1.0$ | 75.16 | 86.92 | 87.10 | 87.01 |
| $\lambda = 1.5$ | 73.91 | 83.02 | 81.76 | 82.39 |
| $\lambda = 2.0$ | 71.34 | 82.15 | 80.10 | 81.11 |

*Table 8.* Sensitivity analysis of hyperparameter $\tau$.

| Values of $\tau$ | $mAP50\%$ | $Pr\%$ | $Re\%$ | $F1\%$ |
|---|---|---|---|---|
| $\tau = 0.1$ | 76.82 | 89.94 | 87.35 | 88.61 |
| $\tau = 0.3$ | 77.65 | 90.21 | 88.74 | 89.47 |
| $\tau = 0.5$ | **78.24** | **90.85** | **90.40** | **90.62** |
| $\tau = 0.7$ | 77.91 | 89.96 | 89.82 | 89.89 |
| $\tau = 1.0$ | 77.38 | 89.31 | 89.05 | 89.18 |

The temperature parameter $\tau$ in the Gumbel–Softmax mechanism is set to 0.5 in our network. A larger temperature produces a smoother probability distribution, which improves training stability but may weaken the discriminability of feature selection. In contrast, a smaller temperature drives the distribution toward discretization, enhancing the sparsity and selectivity of feature representations while potentially introducing optimization instability. Therefore, an appropriate choice of the temperature parameter is critical to model performance. To further investigate the effect of different temperature settings on detection performance, we conduct comparative experiments under multiple values of $\tau$ and comprehensively

evaluate detection accuracy, false alarm rate, and model stability to determine the optimal configuration. The experimental results are presented in Table 8.

For the hyperparameters $\alpha$ and $\beta$ in the loss function of the implicit stream branch, they are set to 1 and 0.1, respectively. The corresponding sensitivity analysis results are reported in Tables 9 and 10.

*Table 9.* Sensitivity analysis of hyperparameter $\alpha$.

| Values of $\alpha$ | $mAP50\%$ | $Pr\%$ | $Re\%$ | $F1\%$ |
|---|---|---|---|---|
| $\alpha = 0.1$ | 76.48 | 88.91 | 87.42 | 88.16 |
| $\alpha = 0.5$ | 77.36 | 89.74 | 88.96 | 89.35 |
| $\alpha = 1.0$ | **78.24** | **90.85** | **90.40** | **90.62** |
| $\alpha = 1.5$ | 77.71 | 89.92 | 89.31 | 89.61 |
| $\alpha = 2.0$ | 76.83 | 88.77 | 88.54 | 88.65 |

*Table 10.* Sensitivity analysis of hyperparameter $\beta$.

| Values of $\beta$ | $mAP50\%$ | $Pr\%$ | $Re\%$ | $F1\%$ |
|---|---|---|---|---|
| $\beta = 0.01$ | 77.02 | 89.31 | 88.24 | 88.77 |
| $\beta = 0.05$ | 77.78 | 90.14 | 89.32 | 89.73 |
| $\beta = 0.1$ | **78.24** | **90.85** | **90.40** | **90.62** |
| $\beta = 0.2$ | 77.59 | 89.88 | 89.57 | 89.72 |
| $\beta = 0.5$ | 76.64 | 88.96 | 88.71 | 88.83 |

### A.3. Comparison on Computational Efficiency

Table 11 compares computational cost and detection performance. Our method achieves the best F1 score of 90.62%, outperforming the second-best method, Tridos, by 3.77%. In summary, existing methods often face a trade-off: some (e.g., TSI-Net) prioritize computational efficiency at the expense of $F1$ score, while others attain higher detection performance but require substantial computational overhead. As a result, CodeMamba outperforms other SOTA methods in detection performance while keeping fast speed and low cost.

*Table 11.* Comparison of Computational Efficiency: model size (Params), floating-point operations (FLOPs), and frames per second (FPS) at an input resolution of $512 \times 512$.

| Methods | Params (M) $\downarrow$ | FLOPs (G) $\downarrow$ | FPS $\uparrow$ | F1 $\uparrow$ |
|---|---|---|---|---|
| TSI-Net (Zhuang et al., 2025) | **6.29** | **23.7** | **29.18** | 82.18 |
| MICPL (Chen et al., 2024b) | 8.84 | 77.20 | 25.56 | 85.34 |
| TMP (Zhu et al., 2024b) | 16.41 | 92.85 | 6.91 | 83.89 |
| MoPKL (Chen et al., 2025) | 9.46 | 119.64 | 10.03 | 86.84 |
| STME (Peng et al., 2025) | 9.85 | 41.92 | 11.95 | 85.96 |
| Tridos (Duan et al., 2024) | 14.13 | 130.72 | 13.71 | 86.85 |
| SSTNet (Chen et al., 2024a) | 11.95 | 123.59 | 9.24 | 85.11 |
| **Ours** | 8.65 | 39.28 | 26.52 | **90.62** |

*Table 12.* The impact of varying the number of input frames on both detection performance and computational cost.

| Input frames | Params (M) $\downarrow$ | FLOPs (G) $\downarrow$ | FPS $\uparrow$ |
|---|---|---|---|
| 5 | 8.65 | **39.28** | **26.52** |
| 10 | 8.65 | 71.84 | 18.94 |
| 15 | 8.65 | 104.37 | 14.76 |

We conduct additional experiments to investigate the impact of varying the number of input frames on both detection performance and computational cost. As shown in Table 12, increasing the number of input frames does not affect the

number of model parameters, since the network architecture remains unchanged. However, the computational cost grows significantly with longer input sequences, leading to increased FLOPs and reduced inference speed.

### A.4. Response Visualizations of Each Module

To further investigate the role of each module in detection, we visualize the feature responses of different modules using heatmaps, as shown in Figure 5. In the raw image (Figure 5(a)) with complex infrared scenarios, small targets tend to be overwhelmed by background noise and structural textures, resulting in insufficient saliency. Through the proposed GBAPM, background regions are significantly suppressed while potential target locations emerge as localized hotspots of high activation, as shown in Figure 5(b). This observation validates that GBAPM effectively learns the background manifold and isolates anomalous regions that deviate from the background statistics. Figure 5(c) shows that by capturing motion singularity, MHSFM results in concentrated responses for small targets, making them more compact and salient. Consequently, the BUWFM is designed to fuse the features extracted from GBAPM and MHSFM to attain optimal feature representations (illustrated in Figure 5(d)), where the targets can be decisively discriminated from background.

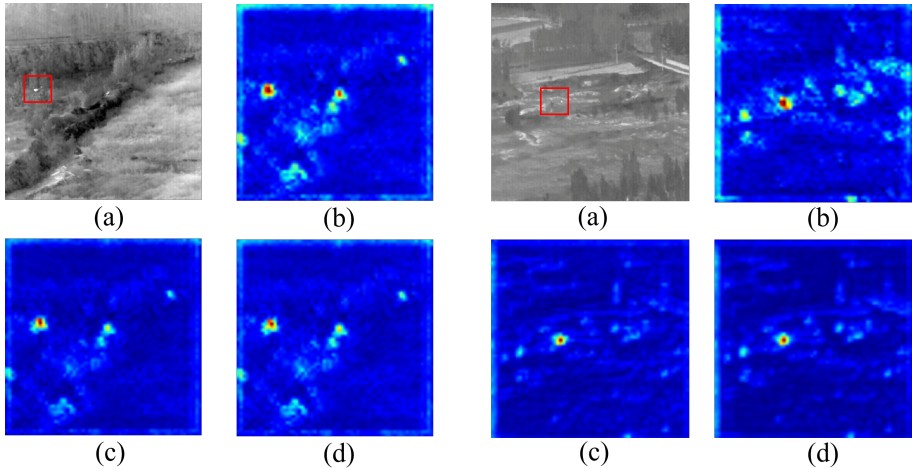

*Figure 5.* Visualizations of heatmap responses from different modules. (a) Raw image. (b) GBAPM. (c) MHSFM. (d) BUWFM.

### A.5. Limitation of Our method

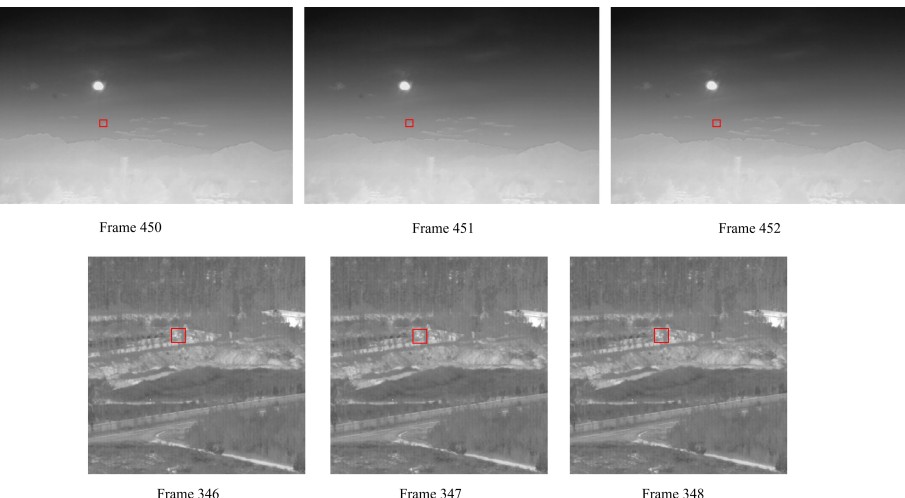

*Figure 6.* Visualizations of scenarios where CodeMamba fails to detect targets.

Based on our analysis, CodeMamba exhibits reduced performance in the following situations: (i) Targets adjacent to high-intensity background clutter. When a target is located near strong background structures (e.g., cloud edges, thermal textures), its weak signal can be partially absorbed into the learned background manifold, leading to missed detections. (ii) Targets with occluded trajectories. In these cases, the temporal consistency assumption of the motion modeling is violated, making it difficult to maintain continuous tracking across frames and occasionally resulting in missed detections. The visualizations of these scenarios are presented in Figure 6.

