# OpenReview forum: "CodeMamba: Shifting from Target Semantics to Self-Supervised Background Manifold Learning for Singularity Detection in Infrared Sequences"
_ICML.cc/2026/Conference — ICML 2026 regular_

### Official Review · Reviewer_exao · 2026-03-06

**Soundness:** 3
**Presentation:** 2
**Significance:** 3
**Originality:** 2
**Overall Recommendation:** 4
**Confidence:** 4

**Summary:**

To solve the  challenge of extreme semantic paucity of targets and representation collapse due to overwhelming class imbalance, this paper propose CodeMamba, which is a collaborative dual-stream framework driven by complementary mechanisms of background manifold modeling and motion singularity capturing.

This framework is motivated by two observations: 1) Background exhibits high spatio-temporal redundancy, and 2) small targets are characterized by localized yet temporally consistent motion.

Specifically, an implicit stream learns background regularity and anomaly localization via a Granularity-Aware Background Anomaly Perception Module (GBAPM), a Motion-Aware Hybrid Spatial-Frequency Mamba Module (MHSFM) is designed to integrates spatial selective scanning with global frequency-domain modeling to capture subtle motion trajectories. and a Bayesian Uncertainty-Weighted Fusion Module (BUWFM) is adopted to adaptively weights each stream. Experiments are conducted on the DAUB and IRDST benchmarks.

**Compliance With Llm Reviewing Policy:**

Affirmed.

**Final Justification:**

The authors have addressed my main concerns, especially regarding the motivation and pseudo-label clarification.

although the authors argue that the method is not simply “heavily engineered,” the architecture still contains multiple complex modules, and the relative necessity of each component could be further clarified beyond standard ablations.

Overall, my main concerns have been largely resolved,  I therefore lean towards a weak accept.

**Key Questions For Authors:**

- Can the authors provide a theoretical or empirical justification for the claims of "Semantic Paucity" and "Representation Collapse" to better motivate their work?

- How does the proposed approach conceptually differ from a learnable, deep extension of Robust PCA? A clear discussion is needed.

- Could the authors provide an ablation study on the DAUB dataset, analogous to Table 2 for IRDST, to demonstrate the generalizability of their modules?

- What is the quality of the pseudo-labels generated by the GBAPM? Is there any analysis  to show that the self-supervision is providing a useful signal?

**Limitations:**

The paper lacks a dedicated discussion of its limitations, particularly an analysis of failure cases. While the paper presents qualitative successes in Figure 3, a rigorous submission should also examine where and why the model fails.

**Strengths And Weaknesses:**

Strengths
- The separation of the problem into background manifold modeling and motion singularity capture is an intuitive perspective.
- The proposed method achieves state-of-the-art on the IRDST dataset, demonstrating its effectiveness in that specific context.

Weaknesses
- The core motivation, namely: "Semantic Paucity within Standard Architectures" and "Representation Collapse due to Extreme Class Imbalance". For a paper at ICML, these claims need to be substantiated, either through a pilot study, theoretical analysis, or at least a more detailed problematization of why existing methods fail.

-  The "two orthogonal inductive biases" bear a strong resemblance to the principles of classical Robust PCA (RPCA), which decomposes data into a low-rank background and a sparse foreground. The paper frames this as a new insight but does not adequately differentiate it.

- "Heavily Engineered" Architecture: The paper introduces three distinct, complex modules (GBAPM, MHSFM, BUWFM).  This  raises the question of whether the performance improvementsi is from the increased architectural complexity and capacity.  The paper would benefit from a clear identification of which component is the most critical and a deeper analysis of why it works, rather than a description of the whole pipeline.

- The implicit stream is described as self-supervised.  But the final pseudo-labels ($P{pseudo}$) are used to gate features from all streams, making the entire pipeline supervised.  Moreover, the paper does not analyze the quality of the generated pseudo-labels or discuss the potential risks of  imperfect self-supervised signals.

- Insufficient ablation validation: The ablation study in Table 2 lacks a clear definition of the baseline.  And all ablations are performed only on the IRDST dataset. There is a risk that the proposed modules are an inductive bias overfit to the specific characteristics of IRDST.  Presenting the same ablation study on the DAUB dataset is needed to demonstrate the effectiveness of the proposed methods.

- The Impact Statement is completely mismatched to the paper's content, mentioning "automated structural health monitoring" which is not the focus of this work on infrared small target detection.

- The overall writing, method description, and contribution have significant room for improvement.  The abstract and introduction contain vague statements. The method section is dense and difficult to follow, often reading like a list of operations rather than a clear, motivated narrative.

---

> ### Author Rebuttal · Authors · 2026-03-31
>
> **Q1&W1. Theoretical or empirical justification for "Semantic Paucity" and "Representation Collapse"**
> We agree that providing further theoretical justification for “Semantic Paucity” and “Representation Collapse” will strengthen the motivation of our work. These two phenomena represent long-standing core challenges in infrared small target detection [1-3].
> **Semantic Paucity:** Infrared small targets typically occupy only a few pixels. Therefore, they lack clear texture, edge, and shape information, making them difficult to recognize using conventional semantic features. Recent surveys [1] have identified “mitigating information loss in deep networks” as a key challenge. It demonstrates that existing methods commonly suffer from insufficient target representations.
> **Representation Collapse:** This phenomenon stems from extreme imbalance between targets and background. Prior studies [2] have demonstrated that reducing target size typically leads to a more severe performance drop than reducing the number of target instances. In the context of MISTD, this imbalance is particularly pronounced. Furthermore, existing works [3] also indicate that models are easily dominated by background patterns while overlooking weak targets. Consequently, small target representations are overwhelmed by dominant background statistics, making representation collapse a critical challenge.
> [1] Infrared Dim Small Target Detection Networks: A Review. Sensors, 2024.
> [2] A systematic study of the foreground-background imbalance problem in deep learning for object detection. arXiv preprint arXiv:2306.16539, 2023.
> [3] BPGA: Enhancing infrared small and dim target detection with background prior guide and vision graph attention. Optics & Laser Technology, 2025.
>
> **Q2&W2. The conceptual differences between the proposed method and learnable deep Robust PCA need to be clearly discussed.**
> Robust PCA models the background as low-rank and targets as sparse perturbations. This linear assumption fails in dynamic infrared scenes with complex disturbances (e.g., drifting clouds, undulating waves), resulting in inaccurate background modeling and target separation.
> Our method models the background as a nonlinear dynamic manifold, replacing the low-rank assumption. We adopt a framework of background manifold modeling guided by anomaly perception, which is better suited to complex, nonlinear, and time-varying infrared scenes. Furthermore, small infrared targets are not merely anomalies in background manifold but also exhibit spatiotemporal motion singularity. Therefore, we introduce MHSFM to complement the implicit background branch, jointly modeling target motion continuity and singularity in spatial and frequency domains.
>
> **Q3. Supplement an ablation study on the DAUB dataset to verify module generalizability.**
> We thank the reviewer for this constructive suggestion. To demonstrate the generalizability of our proposed modules, we have conducted an ablation study on DAUB, as listed in the Table 1(https://anonymous.4open.science/r/CodeMamba-B784).
>
> **Q4&W4. Analysis of the Quality of Pseudo-Labels Generated by GBAPM and the Effectiveness of Self-Supervision**
>
> As shown in original Appendix Figure 4, the reconstruction residual maps produced by GBAPM exhibit high activation precisely at target locations, while effectively suppressing background regions. This indicates that the pseudo-labels provide accurate spatial cues for target localization. To evaluate the quality of pseudo-labels, we provide additional visualizations across diverse challenging scenarios, as shown in Figure 1(https://anonymous.4open.science/r/CodeMamba-B784).
> Additionally, pseudo-labels generated by fixed rules risk propagating initial errors through training, which exacerbates the difficulty in infrared small target detection, where targets are weak with complex background. In contrast, pseudo-labels generated by the self-supervised mechanism are not static but iteratively refined throughout training. The iterative refinement of the background manifold drives continuous improvement of the pseudo-labels, resulting in progressively more precise target localization.
>
> **W3.** We thank the reviewer for this constructive concern. CodeMamba achieves superior performance with only 8.65M parameters and 39.28G FLOPs, which is even more efficient than several strong baselines. Therefore, our performance gains stem from task-specific design rather than mere architectural stacking.
> Among the three modules, we identify GBAPM as the most critical component. To overcome target semantic paucity, this module learns dominant background patterns via PVQ instead of directly learning limited target semantics.
>
> **W6&W7.** We thank the reviewers for pointing out these issues. We have corrected the mismatched impact statement section and comprehensively optimized the writing, logic, and clarity of expression throughout the entire paper.

---

> > ### Author Rebuttal · Reviewer_exao · 2026-04-04
> >
> > Thank you for your response. A few points still need clarification:
> >
> > While I understand the reliance on prior literature, I was hoping to see some empirical evidence from your own experiments (e.g., quantitative analysis of semantic paucity or representation collapse in your specific setting) to further strengthen the motivation.
> > Also, I’m curious about the pseudo-labels. In what scenarios might they fail or not work well?

---

> > > ### Author Response · Authors · 2026-04-07
> > >
> > > We thank the reviewer for these constructive suggestions.
> > > **Empirical evidence for semantic paucity and representation collapse:** We agree that a quantitative analysis can better motivate our method. Following your suggestion, quantitative analyses on our datasets to empirically validate the semantic paucity and representation collapse phenomena are discussed as follows.
> > > **To quantify semantic paucity**, we first adopt a **normalized Fisher Discriminant Ratio** (${\mathrm{FDR}}_{norm}$):
> > >
> > > $$
> > > {{\mathrm{FDR}}_{norm}}=\frac{\mathrm{FDR}}{\mathrm{FDR}+1}∈[0,1)
> > > $$
> > >
> > > where $\mathrm{FDR}$ is defined as the ratio of between-class scatter to within-class scatter between target and background features:
> > >
> > > $$
> > > \mathrm{FDR} = \frac{\mathrm{tr}(S_B)}{\mathrm{tr}(S_W)} = \frac{n_{fg} \||\mu_{fg} - \mu\||^2 + n_{bg} \||\mu_{bg} - \mu\||^2}{\sum\limits_{x \in fg} \||x - \mu_{fg}\||^2 + \sum\limits_{x \in bg} \||x - \mu_{bg}\||^2}
> > > $$
> > >
> > > where $S_B$ and $S_W$ denote the between-class and within-class scatter matrices, respectively. $n_{fg}$ and $n_{bg}$ are the numbers of foreground (target) and background feature samples, respectively. $\mu_{fg}$ and $\mu_{bg}$ denote the corresponding class means, and $\mu$ is the global mean computed over all features.
> > >
> > > $$\mu = \frac{n_{fg}\mu_{fg} + n_{bg}\mu_{bg}}{n_{fg} + n_{bg}}$$
> > >
> > > A value of ${\mathrm{FDR}}_ {norm}$ close to 0 indicates that target and background features are nearly inseparable in the representation space. As listed in table below, the values of ${\mathrm{FDR}}_ {norm}$ on IRDST and DAUB are 0.00566 and 0.01945, respectively. Additionally, target size statistics show that the median target sizes on IRDST and DAUB are only 51.2 and 64.8 pixels, respectively. Together, these quantitative results confirm that images in the datasets contain extremely limited discriminative information for infrared small targets, demonstrating that they indeed suffer from severe semantic paucity in our specific setting.
> > > **To quantify representation collapse**, we adopt the Average Pairwise Cosine Similarity ($\mathrm{APCS}$) as a quantitative metric:
> > >
> > > $$\mathrm{APCS} = \frac{1}{N(N-1)} \sum_{i \neq j} \mathrm{cos\_sim}(\mathbf{z}_i, \mathbf{z}_j)$$
> > >
> > > $\mathrm{APCS}$ measures the average cosine similarity between all pairs of feature vectors within a set, reflecting the degree of crowding or aggregation in the feature space. In an ideal feature space, different classes are well-separated with a low $\mathrm{APCS}$, while same-class samples are compact with a high $\mathrm{APCS}$. For infrared small target detection, an excessively high $\mathrm{APCS}$ indicates severe representation collapse, as the model maps nearly all inputs (small targets and background) into an extremely narrow region of the feature space. The $\mathrm{APCS}$ values on IRDST and DAUB are 0.987 and 0.967, respectively (see table below), quantitatively confirming this collapse.
> > > |Datasets|Median target sizes|${\mathrm{FDR}}_ {norm}$|$\mathrm{APCS}$|
> > > |:---:|:---:|:---:|:---:|
> > > |IRDST|51.2|0.00566|0.987|
> > > |DAUB|64.8|0.01945|0.967|
> > >
> > > As a result, the above quantitative analysis provides strong empirical evidence for the semantic paucity and representation collapse phenomena we identified.
> > >
> > > **Failure cases of $P_{pseudo}$​:** We thank the reviewer for this question. We have analyzed the failure scenarios of the pseudo-label $P_{pseudo}$ where it may underperform: (1) high-intensity static structural clutter (e.g., stationary ground clutter), and (2) approximation of dynamic thermal textures (e.g., cloud clutter). Furthermore, we have shown the two cases in Figure 1 and Figure 2(https://anonymous.4open.science/r/Rebuttal_15984-2B19/).

---

### Official Review · Reviewer_iGMV · 2026-03-10

**Soundness:** 2
**Presentation:** 3
**Significance:** 2
**Originality:** 2
**Overall Recommendation:** 3
**Confidence:** 5

**Summary:**

Since infrared small targets and background clutter lack distinct semantic features, this paper attempts to remedy this issue by exploiting the complementary mechanisms of background manifold modeling and motion singularity capturing, thereby boosting the detection performance of infrared small targets. In this work, an implicit stream and an explicit stream are deployed to focus on different tasks respectively, and Bayesian uncertainty-weighted fusion is designed for performance evaluation. Experimental results verify the effectiveness of the proposed method to a certain extent.

**Compliance With Llm Reviewing Policy:**

Affirmed.

**Final Justification:**

My concerns have been addressed.

**Key Questions For Authors:**

1.	What exactly is the genuine role of spatial pseudo-labels? Whether it essentially provides an attention mechanism or enables the generation of high-precision labels requires further elaboration. As stated previously, if it is claimed to offer reliable labels, additional detailed explanations and analyses should be supplemented.
2.	The configurations of some hyperparameters are ambiguous. For instance, although the temperature hyperparameter in the Implicit stream is briefly mentioned, its specific value is not clearly stated. It remains unclear whether its setting affects the fundamental performance of the model. Similar issues need to be further clarified throughout the entire paper.
3.	While the perspective of background modeling carries a certain degree of novelty, it remains questionable whether this approach alone can effectively discriminate between background clutter and targets. More concrete analyses and visual demonstrations are lacking; merely claiming that infrared small targets lack distinct features is clichéd and inadequate.
4.	Since this is a multi-frame method, the number of images used for fusion merits in-depth discussion. The experiments presented in this paper were conducted solely based on 5 frames of images. Accordingly, whether image fusion with a larger number of frames remains adaptable, and whether it introduces additional burdens such as computational overhead, are issues that deserve further exploration.
5.	The modules such as GBAPM proposed in this paper do not seem to show a clear correlation with Mamba. It appears that the overall logical framework of the paper would remain unchanged even if these modules were replaced with CNN or Transformer. What are the specific advantages and roles of Mamba in this work?

**Limitations:**

The explicit stream correlates features in the spatiotemporal domain, yet there is a lack of targeted and quantitative analysis regarding whether platform motion impacts the model's performance.

**Strengths And Weaknesses:**

Strengths：
The paper contribute some works in originality for infrared small target detection.

1.	This paper shifts the detection paradigm for infrared small targets by adopting self-supervised learning to model the manifold of complex dynamic backgrounds, where targets are defined as singularities deviating from the manifold. The proposed method has the potential to provide valuable insights for researchers in relevant specific fields.
2.	The dual-stream architecture design and BUWFM enable the fusion of multi-source features. Meanwhile, this framework learns the background prototype distribution and captures the motion singularity of targets based on spatiotemporal continuity.

Weaknesses：

1.	It remains to be further elaborated why background manifold learning can enhance the discrimination capability between targets and background clutter. Generally, both targets and background clutter are high-frequency anomalies relative to the background; thus, what is their intrinsic distinguishability? This question cannot be answered solely based on the current results and analyses. Some performance gains may potentially stem from the fusion of multi-frame information.
2.	This paper lacks direct quantitative evaluation (e.g., Intersection over Union (IoU) against ground-truth annotations) and visualization regarding the intrinsic quality of the generated pseudo-labels. The effectiveness of the proposed method is only indirectly verified through the improvement of final detection performance, which leaves a verifiable weakness in the logical reasoning chain. For instance, LESPS[1] is a representative point-label-based weakly supervised learning method for infrared small target detection. When applied to the open-source dataset IST-A[2], it enables the classic model DNANet to achieve even better instance-level evaluation performance than the fully supervised counterpart in the reviewer’s research. However, the pseudo-labels generated by LESPS are completely inconsistent with the actual target distributions, and the performance gain is unexpectedly caused by the erroneous expansion in target segmentation. Furthermore, targeted analyses of the two datasets are insufficient. For example, the label scale in the IRDST[3] dataset is relatively small according to the reviewer's criteria. In summary, although the authors attempt to contribute to pseudo-label-based research, there is a complete lack of data analysis on the quality of pseudo-labels.
3.	The paper lacks an analysis of platform motion. Although GBAPM exploits the differences between consecutive frames, it fails to analyze the impacts and challenges that platform movement—especially drastic motion—poses to the model.
4.	Some of the experiments may lack rationality: The multi-frame method proposed in this paper processes five images simultaneously by default. Therefore, it may be inappropriate/unreasonable to compare its FPS with that of single-frame processing. In such scenarios, the system should focus more on output latency instead. Does the time latency of the proposed algorithm need to be multiplied by a factor of 5 accordingly?
5.	Some analyses need to be further clarified. The roles of the proposed modules rely entirely on the authors’ statements: for example, GBAPM and MHSFM are claimed to respectively extract and fuse complementary information, but this is entirely implicit, without any visualized intermediate results or concrete analyses. Although the method theoretically extracts spatial and spatiotemporal features, it is not convincing that these features are indeed complementary.


[1] Xinyi Ying, Li Liu, Yingqian Wang, Ruojing Li, Nuo Chen, Zaiping Lin, Weidong Sheng, and Shilin Zhou. Mapping degeneration meets label evolution: Learning infrared small target detection with single point supervision. In Proceedings of the IEEE Conference on Computer Vision and Pattern Recognition (CVPR), pages 15528–15538, 2023.

[2] Xu H, Zhong S, Zhang T, et al. Multiscale multilevel residual feature fusion for real-time infrared small target detection[J]. IEEE Transactions on Geoscience and Remote Sensing, 2023, 61: 1-16

[3] H. Sun, J. Bai, F. Yang, and X. Bai, “Receptive-field and direction induced attention network for infrared dim small target detection with a large-scale dataset IRDST,” IEEE Trans. Geosci. Remote Sens., vol. 61, 2023, Art. no. 5000513.

---

> ### Author Rebuttal · Authors · 2026-03-31
>
> **Q1&W2.** The true role of spatial pseudo-labels requires further elaboration.
> We apologize for the confusion caused by the terminology in our original manuscript. The term "pseudo-label" is indeed imprecise and may have led to misunderstanding.
> In GBAPM, $P_{pseudo}$ functions more like an attention prior that represents the probability map of a region deviating from the learned background manifold, as shown in Figure 1(https://anonymous.4open.science/r/CodeMamba-68FC/). Its primary purpose is to guide the network to focus on potential target regions and to reweight multi-stream features in a task-adaptive manner.
> To avoid further confusion, we will rename $P_{pseudo}$ to anomaly attention map and explicitly clarify its functional role as an attention guidance rather than a kind of pseudo labels.
>
> **Q2.** The configurations of some hyperparameters are ambiguous.
> We agree that the description of some hyperparameters in the current version remains insufficiently complete.
> In our implementation, the temperature parameter $\tau$ in the Gumbel-Softmax mechanism is set to 0.5 for all experiments. We have conducted an ablation study to evaluate its impact on model performance, as shown in the table below
>
> |$\tau$ |mAP50|Pr|Re|F1|
> |:---:|:---:|:---:|:---:|:---:|
> |0.1|76.82|89.94|87.35|88.61|
> |0.3|77.65|90.21|88.74|89.47|
> |**0.5**|78.24|90.85|90.40|90.62|
> |0.7|77.91|89.96|89.82|89.89|
> |1.0|77.38|89.31|89.05|89.18|
>
> For the hyperparameters $\alpha$ and $\beta$ in the loss of the implicit flow branch, we set them to 1 and 0.1 respectively, and the sensitivity analysis is shown in the table below.
> |$\alpha$ |mAP50|Pr|Re|F1|
> |:---:|:---:|:---:|:---:|:---:|
> |0.1|76.48|88.91|87.42|88.16|
> |0.5|77.36|89.74|88.96|89.35|
> |**1.0**|78.24|90.85|90.40|90.62|
> |1.5|77.71|89.92|89.31|89.61|
> |2.0|76.83|88.77|88.54|88.65|
>
> |$\beta$ |mAP50|Pr|Re|F1|
> |:---:|:---:|:---:|:---:|:---:|
> |0.01|77.02|89.31|88.24|88.77|
> |0.05|77.78|90.14|89.32|89.73|
> |**0.1**|78.24|90.85|90.40|90.62|
> |0.2|77.59|89.88|89.57|89.72|
> |0.5|76.64|88.96|88.71|88.83|
>
> The total loss is a composite loss we designed. To determine the optimal weighting coefficient  $\lambda$,
> we have provided a sensitivity analysis in original Appendix.
>
> **Q3,W1&W5.** Whether background manifold learning can effectively distinguish targets from​ background clutter requires further analysis and visualization.
> We fully agree that background modeling alone is insufficient to accurately discriminate targets from clutter in all challenging scenarios. This is precisely the motivation behind our dual-stream design. We use GBAPM as an anomaly sieve to highlight regions that deviate from the background manifold, including real targets and random noise clutter. MHSFM then verifies whether these candidate regions exhibit motion patterns consistent with genuine targets.
> Followed by the reviewer’s suggestion, we have illustrated additional heatmap visualizations across various dynamic scenes in Figure 2(https://anonymous.4open.science/r/CodeMamba-68FC/). These examples demonstrate how GBAPM suppresses background while MHSFM selectively amplifies true target motions, jointly improving detection robustness.
>
> **Q4&W4. Efficiency fairness and the impact of input-frame number**
> Following your suggestion, we have conducted additional experiments to investigate the impact of varying the number of input frames on both detection performance and computational cost.
> |Input frames |Params (M) $\downarrow$|FLOPs (G)$\downarrow$|FPS$\uparrow$|
> |:---:|:---:|:---:|:---:|
> |5|8.65|39.28|26.52|
> |10|8.65|71.84|18.94|
> |15|8.65|104.37|14.76|
>
> We also agree with the reviewer that output latency should be given more attention. At present, the performance and efficiency comparisons of our method are conducted in a manner fully consistent with other multi-frame methods. In future work, we will further include comparisons in terms of output latency.
>
> **Q5. The role and necessity of Mamba in the proposed framework**
> We agree that GBAPM is not inherently dependent on Mamba, which focuses on background manifold modeling.
> The specific role of Mamba in our work lies in the MHSFM module. Mamba efficiently captures long-range temporal dependencies and dynamic state transitions across frames, achieving linear complexity which is a favorable trade-off compared to CNNs and Transformers. Therefore, Mamba plays a unique and essential role in our framework.
>
> **W3.**
> We agree that the impact of platform motion, especially drastic movement, is not explicitly analyzed in our manuscript. GBAPM focuses on dominant background statistics, which are stable under moderate motion. However, under extreme platform motion, the assumption of inter-frame stability breaks down, and the differences between consecutive frames increase significantly. In such cases, the MHSFM becomes more critical, as it captures target motion in the spatial-frequency domain with less sensitivity to global variations.

---

> > ### Author Rebuttal · Reviewer_iGMV · 2026-04-03
> >
> > Thanks for the rebuttal. My concerns have been addressed.

---

> > > ### Author Response · Authors · 2026-04-07
> > >
> > > Thank you for taking the time to review our rebuttal and for confirming that your concerns have been addressed. We sincerely appreciate your constructive comments, which have helped us improve our work.

---

### Official Review · Reviewer_AP3H · 2026-03-12

**Soundness:** 3
**Presentation:** 3
**Significance:** 3
**Originality:** 2
**Overall Recommendation:** 4
**Confidence:** 4

**Summary:**

This paper proposes a combined framework of anomaly detection, spatial frequency analysis and uncertainty-aware feature fusion for infrared small target detection from sequences. For the anomaly detection branch, it utilised Probabilistic Vector Quantization (PVQ) to identify target regions. The motion-aware mamba module is to capture motion singularities of infrared small targets. Feature fusion is via uncertainty-aware weighted sum. Experiments as well as ablation studies shows the combination of these modules are effective.

**Compliance With Llm Reviewing Policy:**

Affirmed.

**Final Justification:**

Authors have resolved my concerns through rebuttal. I keep my score at 4 but not higher, mainly because its originality is not significant.

**Key Questions For Authors:**

Q1. In 3.2, it claims that GBAPM “.... it encounters significant challenges when targets exhibit extremely low intensity or thermal distributions that closely resemble the local background. In addition, the severe data imbalance leads to representation collapse and feature drift, particularly in dynamic scenes.”, it would be useful to see the failure cases of GBAPM. In the appendix, Figure 4 shows the intermediate result of GBAPM response; it would be useful to see more results on more sequences with different background types and clutter to support the claim in 3.2.

Q2. This paper used PVQ; what is the motivation behind it, and how does it compare to for example, a continuous latent space?

Q3. What is the limitation of this method, and for the future work, please provide more details on cross-modal collaborative tasks. How would the proposed framework be useful?

**Limitations:**

This paper doesn't discuss the limitations of this method, or the possible societal impact of this work. The mentioned structural health monitoring does not seem very relevant.

**Strengths And Weaknesses:**

**Soundness**

This paper is technically sound, for both methodology and experiments. The modular approach of anomaly detection via Probabilistic Vector Quantization (PVQ), motion-aware Mamba and uncertainty-aware feature fusion are clear and well combined.

However, some claims are loose. The claim on this method being a new paradigm of modelling background as a complex neural dynamical system is exaggerated; the GBAPM is more close to a self-supervised anomaly detection module, and MHSFM is for spatial-frequency analysis. Though this paper’s approach is proved to be effective in experiments, the workflow is not a paradigm change in infrared small target detection from sequential data.

**Presentation**

This paper is well structured and easy to follow. Diagrams are clear and effective.

**Significance** and **Originality**

This paper contributes to the small target detection problem in infrared sequences, proposing a new combined approach of anomaly detection (treat small targets as anomalies), spatial and frequency analysis, and uncertainty-aware feature fusion. Authors found a combination that works well for this problem, which is practically useful. But the originality of this paper is not significant.

---

> ### Author Rebuttal · Authors · 2026-03-30
>
> **Q1.** More failure cases of GBAPM under diverse background types and clutter conditions
>
> We thank the reviewer for the careful review and constructive suggestions. We fully agree that providing more visualization of GBAPM's failure cases can help substantiate the limitations discussed in Section 3.2.
> In the original Appendix, Figure 4 demonstrates the effectiveness of GBAPM in suppressing background and highlighting anomalous regions, but it does not sufficiently illustrate the challenges that this module faces in dynamic and complex scenes. To address this, we have supplemented additional failure cases across various representative challenging scenarios, including:
> (1)Cloud clutter: dynamic thermal textures introduce strong structural interference;
> (2)Stationary ground clutter: high-intensity, static structures are easily mistaken for targets;
> (3)Dynamic textured backgrounds: temporal variations in background patterns disrupt the manifold assumption.
> These examples, shown in Figure 1(https://anonymous.4open.science/r/CodeMamba-68FC/), provide a more comprehensive and objective view of GBAPM's limitations, aligning with the discussion in Section 3.2. We believe this addition strengthens the transparency and rigor of our analysis.
>
> **Q2.** Motivation of PVQ and its comparison with a continuous latent space
> We thank the reviewer for this insightful question.
> **Motivation:** The core motivation behind PVQ is that although infrared backgrounds are complex, they exhibit strong spatio-temporal redundancy, making them naturally conducive to compact representation using a set of learnable discrete prototypes.
> **Comparison on a continuous latent space:** The continuous latent space offers greater representational flexibility by compressing inputs via continuous latent variables. This flexibility also makes it prone to fitting background, weak targets, and even local noise simultaneously. However, for the MISTD task, this way will obscure the discriminability between background and targets, leading to weak targets to be wrongly absorbed into the background representation.
> **Advantages of PVQ:** In contrast, PVQ offers three key advantages:
> (1)Compact background representation. The discrete codebook forces a compact encoding of the dominant background modes, making target anomaly more salient as deviations from the background manifold.
> (2)Probabilistic quantization. Unlike hard vector quantization, our probabilistic formulation ensures stable optimization and avoids boundary artifacts when modeling continuous thermal fields.
> (3)Hierarchical modeling. The coarse-to-fine codebook design captures both low-frequency background structures and fine-grained local details, facilitating comprehensive background modeling.
>
> **Q3.** Limitations of the method and future directions for cross-modal collaborative tasks.
>
> We thank the reviewer for this constructive suggestion.
> **Limitations.** A key limitation is that our method assumes the target manifests as a discernible anomaly, either within the background manifold or in its motion trajectory. When a target closely resembles the background across thermal, structural, and motion domains, detection becomes inherently difficult.
> **Further work on cross-modal collaborative tasks.** Our framework could be extended to cross-modal collaborative tasks such as infrared-visible fusion or multispectral detection. The GBAPM can learn a shared background manifold across modalities, allowing for unified anomaly detection without task-specific retraining. Meanwhile, the BUAFM can dynamically weight each modality based on its estimated reliability, allowing the model to automatically favor high-quality channels while suppressing noisy ones.

---

> > ### Author Rebuttal · Reviewer_AP3H · 2026-04-05
> >
> > Thanks to the authors for their rebuttal. My key questions are resolved. My comment on the soundness of the exaggerated claim of a new paradigm has not been discussed in the rebuttal. I don't have follow-up questions and keep my score.

---

> > > ### Author Response · Authors · 2026-04-07
> > >
> > > We thank you for confirming that your key questions have been resolved, and we greatly appreciate your time and effort in reviewing our work.
> > > Regarding the concern about the new paradigm claim, we apologize that we did not directly address this point in our previous rebuttal. We understand that the term "new paradigm" may be an overstatement. Our intention is to emphasize a conceptual shift that moves from directly learning discriminative target features to modeling the background manifold as a dynamic component (rather than a static low-rank component), while also incorporating target motion characteristics. Following your reminder, we recognize that **a new perspective or a reformulation** would be more accurate than a new paradigm.
> > > We will revise the manuscript accordingly with more precise language throughout the paper.

---

### Official Review · Reviewer_Joex · 2026-03-13

**Soundness:** 3
**Presentation:** 4
**Significance:** 3
**Originality:** 3
**Overall Recommendation:** 4
**Confidence:** 5

**Summary:**

Moving Infrared Small Target Detection (MISTD) represents one of the most challenging anomaly detection tasks.Mainstream strategies including 3D convolutions, RNNs, and optical flow methods have limited ability to capture long-range dependencies or handle complex background textures effectively. The paper proposes CodeMamba, a collaborative dual-stream architecture reframing MISTD as background manifold modeling and motion singularity capturing. The framework includes three key components. First, the Granularity-Aware Background Anomaly Perception Module (GBAPM) learns background regularity using hierarchical Probabilistic Vector Quantization with coarse-to-fine codebooks. Second, the Motion-Aware Hybrid Spatial-Frequency Mamba Module (MHSFM) integrates spatial and frequency branches with bidirectional scanning to capture motion singularities. Third, the Bayesian Uncertainty-Weighted Fusion Module (BUWFM) adaptively calibrates contribution from each stream by modeling features as Gaussian distributions and computing observation noise variance.Evaluated on IRDST and DAUB datasets, CodeMamba  outperforms all nine compared methods on IRDST.

**Compliance With Llm Reviewing Policy:**

Affirmed.

**Final Justification:**

I'm satisfied with the responses to my questions. Considering the generalization of the method, I think my original score for the paper is appropriate.

**Key Questions For Authors:**

1.Why is there a 19% mAP gap between IRDST and DAUB? Does the method generalize poorly to certain types of infrared scenes?

2.Some recent baseline methods should be compared.

3.What is the theoretical basis for choosing K=512?

4.What types of targets does the method fail to detect?

**Limitations:**

Performance degrades significantly when targets exhibit extremely low intensity or thermal distributions resembling local backgrounds.

**Strengths And Weaknesses:**

Strength:

1.The paper gives an innovative problem formulation. It reframes MISTD as background manifold modeling and motion singularity capturing, providing a principled dual-stream framework. Reframes MISTD as background manifold modeling and motion singularity capturing, providing a principled dual-stream framework.

2.The proposed method achieves state-of-the-art results on both datasets with significant improvements in precision and recall over baselines.

Weakness:

1.Some important infrared sequence small target detection works are omitted, such as [1] Liu et al. IR-MPE: A Long-Term Optical Flow-Based Motion Pattern Extractor for Infrared Small Dim Targets. IEEETransactions on Instrumentation and Measurement,2025. [2] A Global Spatial–Temporal Detection Framework for Infrared Small Targets in Complex Ground Scenes [J]. IEEE Transactions on Geoscience and Remote Sensing,2025.

2.The large gap between IRDST (78.24% mAP50) and DAUB (97.42% mAP50) raises concerns about generalization across different scenarios.

---

> ### Author Rebuttal · Authors · 2026-03-30
>
> **Q1&W2.** What causes the performance gap between IRDST and DAUB?
>
> We thank the reviewer for this insightful observation. The performance gap between IRDST and DAUB is a critical phenomenon that deserves careful analysis. After examining the dataset characteristics, we find that IRDST presents a more challenging scenario due to two main factors: (1) the presence of more complex and dynamic background clutter, and (2) targets are often adjacent to high-intensity, easily confusable background structures. These factors collectively increase the difficulty of reliable target detection on IRDST.
> As noted by the reviewer, our method exhibits reduced detection performance for targets adjacent to high-intensity, easily confusable background regions, as shown in Figure 1(https://anonymous.4open.science/r/CodeMamba-68FC/).
>
> **Q2&W1.** Some recent baseline methods should be compared.
>
> We thank the reviewer for this valuable suggestion. Following your recommendation, we have conducted additional experiments to compare our method with two more recent baseline methods: GST-Det (2025)[1] and MIST-Net (2026)[2]. Since the codes for these methods are not publicly available, we directly report the results as presented in their respective papers, using the same datasets as ours to ensure a fair comparison. As listed in the table below, our method consistently outperforms both baselines across all metrics.
>
> **Supplementary Comparison with Recent Baseline Methods**
> |Method|DAUB|DAUB|IRDST|IRDST|
> |:---:|:---:|:---:|:---:|:---:|
> | |mAP50|F1|mAP50|F1|
> |MIST-Net|95.62|97.30|73.49|87.00|
> |GST-Det|97.31|-|-|-|
> |**Ours**|97.42|99.42|78.24|90.62|
>
> In addition, we also attempt to include IR-MPE (2025) [3], but find that a fair comparison is infeasible due to the lack of public code and mismatched datasets.
> [1]	A Global Spatial-Temporal Detection Framework for Infrared Small Targets in Complex Ground Scenes. IEEE Transactions on Geoscience and Remote Sensing, 2025.
> [2]	Moving Infrared Small Target Detection via Motion-aided Integrated Spatial-Temporal Network. IEEE Transactions on Geoscience and Remote Sensing, 2026.
> [3]	IR-MPE: A Long-Term Optical Flow-Based Motion Pattern Extractor for Infrared Small Dim Targets. IEEE Transactions on Instrumentation and Measurement, 2025.
>
> **Q3.** What is the theoretical basis for choosing K=512?
>
> We thank the reviewer for this insightful question. In GBAPM, K denotes the codebook size, which directly determines the effective number of samples assigned to each codeword $N_{l,k}^{(t)}$ via Eq. (1) and (2)
> $$
> \pi_ {c, k} = \frac{\exp\left(-\left( \||z - e_{c, k}\||_ 2^2 + g_ k\right)/\tau\right)}{\sum_ {k} \exp\left(-\left( \||z - e_{c, k}\||_2^2 + g_k\right)/\tau\right)}\tag{1}
> $$
>
> $$
> N_ {l,k}^{(t)} = \gamma N_ {l,k}^{(t-1)} + (1-\gamma) \sum_ {i=1}^{H\times W} \pi_ {l, k, i}\tag{2}
> $$
>
> Taking the coarse codebook as an example, when K is too small, the codebook lacks sufficient capacity to represent diverse background patterns. In this case, the denominator in Eq. (1) sums over a limited number of candidates, resulting in higher assignment probabilities and thus more samples per codeword via Eq. (2). Consequently, distinct background modes are forced to share the same codewords, leading to ambiguity and hindering accurate background reconstruction. Conversely, when K is too large, the denominator sums over significantly more candidates, reducing the effective samples per codeword. As a result, each codeword receives insufficient samples to reliably capture its corresponding background mode. As shown in Appendix Table 5, our method achieves the best performance when the codebook size is set to K=512.
>
> **Q4.** What types of targets does the method fail to detect?
>
> We thank the reviewer for this important question. Understanding the limitations of our method is crucial for a fair assessment and for guiding future improvements.
> Based on our analysis, CodeMamba primarily fails to detect targets in the following scenarios:
> (1)	Targets adjacent to high-intensity background clutter, as illustrated in Figure 2(https://anonymous.4open.science/r/CodeMamba-68FC/). When a target is located in close proximity to strong background structures (e.g., cloud edges, thermal textures), its weak signal can be partially absorbed into the learned background manifold, leading to missed detections.
> (2)	Targets whose trajectories are occluded, as shown in Figure 2(https://anonymous.4open.science/r/CodeMamba-68FC/). In these cases, the temporal consistency assumption of our motion modeling is violated, making it difficult to maintain continuous tracking across frames and occasionally resulting in missed detections.

---

> > ### Author Rebuttal · Reviewer_Joex · 2026-04-03
> >
> > I'm satisfied with the responses to my questions. Considering the generalization of the method, I think my original score for the paper is appropriate.

---

> > > ### Author Response · Authors · 2026-04-07
> > >
> > > We sincerely thank the reviewer for the careful evaluation and for confirming that the main concerns have been satisfactorily addressed. We greatly appreciate your constructive feedback, which has helped us improve the clarity and rigor of the manuscript.

---

### Decision · Program_Chairs · 2026-04-30

**Decision:**

Accept (regular)

**Comment:**

All reviewers are in favor of acceptance. Although iGMV had a final score of weak reject, they acknowledged that the rebuttal addressed all of their concerns. The other reviewers all provided weak accept ratings, praising the paper for its innovative approach in modeling the complex dynamics of the background, and its experimental methodology. The authors should incorporate the reviewers suggestions in the final version.